# Exploring semantic information in disease: Simple Data Augmentation Techniques for Chinese Disease Normalization

## Abstract

The disease is a core concept in the medical field, and the task of normalizing disease names is the basis of all disease-related tasks. However, due to the multi-axis and multi-grain nature of disease names, incorrect information is often injected and harms the performance when using general text data augmentation techniques. To address the above problem, we propose a set of data augmentation techniques that work together as an augmented training task for disease normalization. Our data augmentation methods are based on both the clinical disease corpus and standard disease corpus derived from ICD-10 coding. Extensive experiments are conducted to show the effectiveness of our proposed methods. The results demonstrate that our methods can have up to 3% performance gain compared to non-augmented counterparts, and they can work even better on smaller datasets.

## 1 Introduction

The disease is a central concept in medical text processing problems. One of the most important tasks, i.e. disease normalization, uses diseases as both input and output to match the diagnoses terms used in clinical documents to standard names in ICD coding. The disease normalization task mainly faces the following three challenges. First, different writing styles. The writing styles of the diseases can be diversified, where different doctors have different writing habits, so a single disease might result in thousands of versions of names. Second, data scarcity, where some diseases may not be covered in the training set, which often leads to few-shot or zero-shot scenarios. For example, in the Chinese disease normalization dataset CHIP-CDN, there are 40472 diseases to classify, but only data of 3505 diseases (i.e. less than 10% of all diseases) are provided in the training set. Figure 1 illustrates the data scarcity problem in CHIP-CDN dataset. Third, semantics density. The length of disease names is usually short, which makes every character carries huge semantic information. The meanings of the diseases are very different from each other even if they share a lot of common characters, and a single change in characters could result in dramatic change in semantic meaning. For instance, "髂总动脉夹层 (Common iliac artery dissection)" and "劲总动脉夹层 (Common carotid artery dissection)" are only different in one character, but the positions of those diseases are very distinct, from the upper half of the body part to the lower half.

Among all the challenges we discussed, data scarcity is the biggest one, since other problems usually can be solved by providing larger datasets for models to learn. A common way to address the data scarcity problem is through data augmentation. There are numerous data augmentation methods for general corpora such as synonym replacement or back translation. Wei & Zou (2019) has shown that simple text data augmentation methods can be effective for text classification problems. However, because of the unique structure of disease names (i.e. semantics density), general text data augmentation methods do not work well on them, and sometimes even hurt the overall performance. For example, if random deletion Wei & Zou (2019) is performed on disease "阻塞性睡眠呼吸暂停 (Obstructive Sleep Apnoea)" and results in "阻塞性睡眠 (Obstructive Sleep)", that would dramatically change the meaning of that disease name and makes it become another disease. Admittedly, general

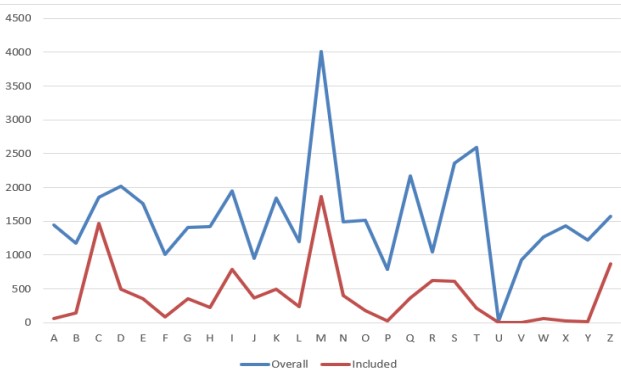

Figure 1: Data scarcity problem in CHIP-CDN dataset. The blue line represents the overall amount of diseases in ICD coding classified by the first coding letter, and the red line represents the number of diseases provided by the CHIP-CDN training set.

data augmentation methods may be able to address the challenge of different writing styles, as performing random operations on texts can be seen as a way to emulate different writing behaviors. However, due to the above reasons, general data augmentation methods tend to hurt performance, which is demonstrated in our experiments. Therefore, designing data augmentation methods specific to disease corpus is necessary. To bridge this gap, we propose a set of disease-oriented data augmentation methods to address this problem.

As with other disease-related tasks, disease normalization can be thought as a process of text matching, from clinical names to standard names in ICD coding. Therefore, the key to this task is for the model to learn great encoding that contains enough similar information for each disease. For instance, the model needs to tell that ”左肾发育不全 (Left renal agenesis)” and ”先天性肾发育不全 (Congenital renal agenesis)” are the same disease while ”髂总动脉夹层 (Common iliac artery dissection)” and ”颈总动脉夹层 (Common carotid artery dissection)” are not, despite that they both share a lot of common characters.

Our methods are based on the following two assumptions. First, disease names have the property of structural invariance. A disease name consists of several different types of key elements, such as location, clinical manifestations, etiology, pathology, etc. In the pair of clinical disease and standard ICD disease, the specified elements can correspond in most cases. Therefore, we can replace a specific element between the pair of clinical disease and standard ICD disease at the same time to generate new pairs. The matching relationship of the newly generated clinical disease and the ICD standard disease pairs can still be maintained. We screened the generated standard ICD diseases to ensure that they belonged to the correct label and that the pairs are effective. It should be noticed that replacing components could derive a new clinical disease name that turns out to be fake (i.e. the disease actually does not exist), but the key point here is to make models learn the necessary semantic association within the diseases. Second, labels in the disease normalization task have transitivity properties. In specific, a more specified description of an object can be comprised into a larger group where the descriptions are more coarse, e.g. a yellow chair is also a chair. In the ICD coding system, there are also different and clear granularities of diseases. Therefore, we can treat the fine-grained disease as their coarse-grained upper disease by assigning them father labels.

Normally, a data augmentation method generates new data and trains them along with the existing data, without altering the training paradigm. However, the disease normalization task assigns each disease a unique label, while our methods augment the labels. Therefore, if the traditional training paradigm is still applied to our augmentation methods, a same input disease in the dataset may get different labels, which will make the model difficult to train due to label confusion. To overcome this problem, we treat the data augmentation operation as a pre-training task (we call it augmented training) prior to the original task, so that the model can first learn the necessary semantic information within diseases and then leverage that information when fine-tuning on the actual normalization dataset.

Additionally, both unnormalized disease names from the tasks and standard ICD names of the diseases can be used as inputs in the data augmentation process. A unique advantage of using standard ICD names to perform data augmentation as a pre-training task is that the model can get the whole picture of the disease-related information from ICD coding, which includes all classes of diseases, even before the actual training of the downstream task. Therefore, with all those information injected, the model can perform much stronger on smaller datasets where lots of class labels are not able to be seen in the training set.

To the best of our knowledge, we are the first to explore the semantic components and information within disease names. We believe the research on disease name enhancement has high research value and can benefit various downstream tasks. To summarize our contributions:

- We propose a set of data augmentation methods for the Chinese disease normalization tasks.

- Experiments validate that general data augmentation methods have the potential to impair the disease normalization task. However, our method has obvious performance gain on the task based on various baseline models.

- We also analyze the reasons why the proposed method is effective.

## 2   Background

ICD coding. ICD, the acronym of the International Classification of Diseases, is an international unified classification of diseases developed by the World Health Organization, and ICD-10 is the 10th version of ICD coding which is used in our work. The coding is a combination of letters and numbers, which classifies diseases according to their etiology, pathology, clinical manifestations, and anatomical locations, so that they form a hierarchical coding structure. ICD also adopts a multi-grain fashion where coarse-grained disease are followed by fine-grained diseases.

Disease normalization task. In clinical practice, doctors will fill in the name of the disease according to clinical diagnosis standards along with their own writing habits, which makes a single disease name hundreds of versions. The disease normalization task is to match disease names written in different styles into a single standard name provided by ICD coding. After the disease normalization process, researchers can perform further operations upon the normalized names to realize all kinds of functions used in wise medical applications. The task can be formalized into the following operation: X -> Y, where X represents the clinical disease names and Y represents the standard ICD names.

NER. NER stands for Named Entity Recognition, which is a common task in Natural Language Processing. It aims to identify entities that have practical values and their locations from unstructured texts. The classification of these entities may include persons, organizations, locations, etc. In this work, we use an NER tool trained by ourselves to identify elements in disease names in order to perform data augmentation. Additionally, we argue that any NER tool that can identify elements in disease names should be fine, and our work mainly focus on the data augmentation methods.

## 3   Related Work

In this section, we first introduce related works of data augmentation, then we introduce medical data-driven research works that are similar to ours.

### 3.1   Data Augmentation

Data augmentation is a technology to synthesize new data based on existing data as a way to expand the amount of dataset. It is often used when the amount of data is not enough, and it can also act as a regularizer to prevent the model from overfitting the training set.

Unlike images, where it is relatively easy to augment data as well as keep the semantic information intact, data augmentation in texts is more difficult, due to its unstructured form Ng et al. (2020). Many works focus on augmentations directly on the input: Wei & Zou (2019) propose four simple augmentation methods base on character-level noise injection, which are replacement, insertion, swap, and deletion. Their methods are quite straightaway and effective, but the augmentation results may cause unwanted noise by not following the grammar rules. Back translation, augments data by translating the original text to a second language and then translating it back. This method can keep the semantic meaning well of the original text, but the augmented results are lack of diversity and sometimes restricted by the translation tool. In order to make the augmented data more realistic, Kim et al. (2022) leverages lexicalized probabilistic context-free grammars to capture the intricate compositional structure of natural language and then perform word replacements. This method yields good results, but grammar-based methods for general text are difficult to generalize to specialized areas, such as medicine.

There are also methods that leverage pre-trained language models to perform data augmentation. Ng et al. (2020) use MLM objective in BERT Devlin et al. (2018) to mask out some words and then regenerate it. Wu et al. (2019) also uses MLM task as well as changing the segment ids to class labels. Kumar et al. (2020) compares three kinds of data augmentation methods using a conditional pre-trained model, namely auto-encoder, auto-regressive, and seq2seq. A problem with these methods is that the semantic meaning of the original sentence may change after several MLM replacements.

Semi-supervised learning can also be a way to perform data augmentation by leveraging the vast amount of unlabeled data. Berthelot et al. (2019) uses MixUp to guess the low-entropy labels of the augmented data and then mixes the labeled and unlabeled data to derive a loss term, and Xie et al. (2020) performs data augmentation on unlabeled data for consistency training. However, we only focus on augmenting the data itself instead of semi-supervised learning objectives in this work.

### 3.2 Data approaches on medical data

While most researches focus on the effect of data augmentation on general text data, there are also works that try to explore the possibility of data augmentation operations on medical text data. In this section, we mainly introduce data augmentation on medical text data and other related research works.

There are works that focus on the synonym replacement in medical terms. Falis et al. (2022) and Abdollahi et al. (2021) leverage Unified Medical Language System (UMLS) to find medical synonyms to perform replacements after certain medical terms are identified in classification texts. Focusing on the ICD-coding task, Falis et al. (2022) also replaces both the medical terms in raw texts and the classification label to get new training data. While their works mainly focus on replacing the whole medical term, we investigate the possibility of replacing the components of the medical terms by exploring the semantic structures within them.

Additionally, Ansari et al. (2021) investigates the performance of EDA, conditional pre-trained language models and back translation to perform data augmentation on social media texts for mental health classification. Wang et al. (2020a) proposes Segment Reordering as a data augmentation technique to keep the medical semantic meaning intact. Wang et al. (2020b) use pre-trained language models fine-tuned on General Semantic Textual Similarity (STS-G) data to generate pseudo-labels on medical STS data, and then perform iterative training.

## 4 Methods

In this section, we introduce the details of our proposed data augmentation methods and the overall pipeline. Since the significance of data augmentation is to inject the model with extra knowledge, the key point is to explore the components and relations in diseases so that the model can have a broad sense of the internal structures of the diseases. Therefore,

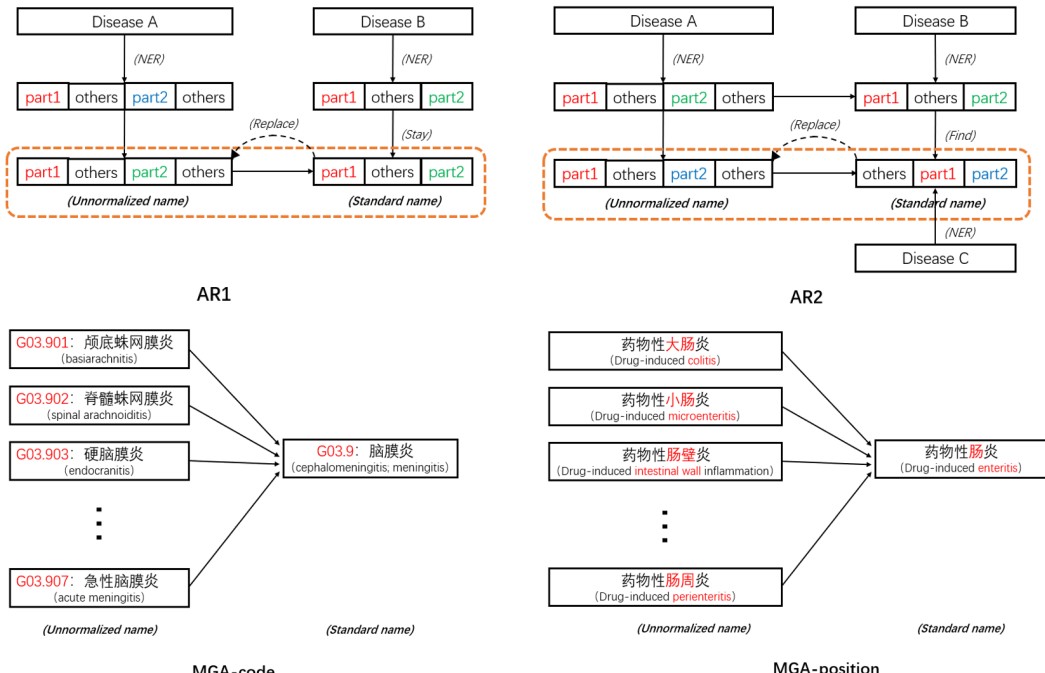

Figure 2: Illustration of our proposed data augmentation methods. The upper half of the figures is the illustration of Axis replacement methods, and the lower half is the illutration of Multi-Grain Aggregation.

we leverage the multi-axis and multi-grain nature of the diseases to design all of the data augmentation methods.

First of all, the disease names are composed of several elements, which include but are not limited to etiology, pathology, clinical manifestations, anatomical location, chronicity, degree type, etc. For ease of expression, we merge and select from all those elements into three main categories, which are disease center, anatomical location and disease quality. This shows the multi-axis nature of the diseases.

- Disease Center: Disease center, which may include etiology and pathology, is the minimal word that describes the nature of a disease. It defines the main category of a disease, such as "disorders" for "Other disorders of the eye with mcc".

- Anatomical Location: Anatomical Location is a part of the human body that have actual meaning in anatomy. It indicates which part of the human body is ill.

- Disease Quality: The quality of a disease which indicates the subtype of the disease, such as "Drug-induced" for "Drug-induced peripheral neuropathy".

With these three axis words, all kinds of disease names can be combined by them.

Second, a disease can be described by multiple granularities. An upper disease is a coarse-defined disease and a lower disease is a fine-grained disease. The ICD coding contains lots of upper-lower disease pairs by assigning them different lengths of code. For example, in "ICD-10 Beijing Clinical Version 601", the disease name of code "A18.2" is "外周结核性淋巴结炎 (Peripheral Tuberculous Lymphadenitis)" and "A18.201" is "腹股沟淋巴结结核 (Inguinal lymph node tuberculosis)". "Peripheral Tuberculous Lymphadenitis" is a coarse-defined disease due to not specifying a single anatomical location. Additionally, a coarse-defined disease can contain multiple fine-grained diseases in ICD coding.

In our intuition, although the disease can only be called the same if all of its components are the same, it is necessary for the model to learn which diseases are more similar than others. Therefore, we define the following data augmentation methods.

4.1 Data Augmentation

We perform data augmentation by assigning pseudo-labels to diseases to describe their relationships so that they can form a new pair of diseases, and we use those pairs to perform augmented training of disease normalization tasks. We divide our methods into two main categories: Axis-word Replacement and Multi-grain Aggregation. We call our proposed disease name data augmentation method DDA. Figure 2 illustrates the overall pipeline of our methods.

Axis-word Replacement (AR): We assume that disease names have the property of structural invariance, which means a name derived by replacing an axis-word in a disease to another one with the same type also makes sense. Since there are often matches of Axis-words between an unnormalized-standard disease pair in the disease normalization task, replacing the corresponding Axis-word in the clinical name with the standard name in the pair at the same time can ensure that the newly-generated pair will still match. To locate all axis-word in the disease, we leverage a Named Entity Recognition (NER) tool trained by ourselves[1]. The entity type includes but is not limited to disease center, anatomical location, and disease quality. We note that the NER tool is just for the use of locating axis-words, and it can be replaced by any modules that can achieve the same function.

We leverage both the ICD-coding and the disease normalization training set to perform axis-word replacement. The detailed descriptions of each category of axis-word replacements are as follows:

- AR1: AR1 is illustrated in the top left corner of Figure 2. First, select a pair of diseases (disease A and disease B) that shares one or more axis (part1 in figure) but is different in another axis (part 2 in figure). Then, replace the part 2 in disease A to be the same part2 in disease B. (Note: disease A can be chosen from any sources, but disease B can only be chosen from the standard ICD-coding list as it serves as the label of a disease normalization pair.)
  - AR1-posotion: Perform AR1 by fixing the disease center and replacing the anatomical location.
  - AR1-center: Perform AR1 by fixing the anatomical location and replacing the disease center.
  - AR1-quality: Perform AR1 by fixing both the disease center and the anatomical location and replacing the disease quality.
- AR2: AR2 is illustrated in the top right corner of Figure 2. First, select a pair of unnormalized-standard diseases from the disease normalization training set. Let the unnormalized disease be disease A, and the standard disease be disease B. Then, find disease C from ICD-coding list that shares one or more axis (part1) but is different in another axis (part2). Finally, replace part2 in disease A to be the same part2 in disease C, so that the replaced disease A and disease C can form a new disease normalization pair.
  - AR2-position: Perform AR2 by fixing the disease center and replacing the anatomical location.
  - AR2-center: Perform AR2 by fixing the anatomical location and replacing the disease center.
  - AR2-quality: Perform AR2 by fixing both the disease center and the anatomical location and replacing the disease quality.

Multi-Grain Aggregation (MGA): We assume that labels in the disease normalization task have transitivity properties. In specific, a more specified description of an object can be comprised into a larger group where the descriptions are more coarse. In the ICD coding system, there are also clear granularities of diseases. The maximum length of code that can be shared between hospitals is 6, and the multi-grain structure contains 3-digit, 4-digit, and 6-digit codes. We observe that the semantic meaning between diseases that share the first

---

[1]We will open source the code of our experiment along with the NER tool for disease names on Github.

3-digit code but are different in the 4th-digit code can be quite different, but the meaning would be a lot similar if the diseases share the first 4-digit code. Therefore, We implement MGA augmentation using the following method.

- MGA-code: we leverage the multi-grain nature of the ICD coding by assigning the label of a 6-digit disease to its corresponding 4-digit disease. We call the method "aggregation" because normally a 4-digit disease can be matched to several 6-digit diseases, so the model can learn which diseases are similar. MGA-code is illustrated in the left bottom of Figure 2.
  - MGA-code1: The 6-digit diseases are directly derived from the ICD-coding list.
  - MGA-code2: The 6-digit diseases are derived from the diseases in CHIP-CDN training set whose labels are a 6-digit ICD disease.
- MGA-position: Apart from the ICD coding, anatomical locations also follow a hierarchical structure, where several smaller positions can be grouped together to form a larger position. Thus, we search for diseases in ICD coding that share the same center and one position is the upper position of another one, and we grouped the classification labels of the lower position diseases to their upper position diseases. MGA-position is illustrated in the right bottom of Figure 2. (Note: the upper position diseases must come from the standard ICD-coding list.)
  - MGA-position1: The lower position diseases are directly derived from the ICD-coding list.
  - MGA-position2: The lower position diseases are derived from the diseases in CHIP-CDN training set.

(Note: In the human body, we call a location the upper position to another position if that location covers a larger area than another. In order to find the upper or lower positions of a position, we construct a position tree document where the anatomical positions in the human body are organized into a tree data structure. We use the constructed position tree to recognize the upper and lower relations above. The same goal can be achieved with other sources containing knowledge bases of human anatomy.)

### 4.2 Training Process

- Taking the augmented data to train the disease normalization task.
- Fine-tuning the original disease normalization dataset.

## 5 Experiments

### 5.1 Dataset

We evaluate the effectiveness of our data augmentation methods on a Chinese disease normalization dataset called CHIP-CDN. CHIP-CDN originates in the CHIP-2019 competition and was collected in A Chinese Biomedical Language Understanding Evaluation Benchmark called CBLUE Zhang et al. (2021). The dataset contains 6000 unnormalized-standard disease pairs in the training set, 1000 pairs in the dev set, and 2000 pairs in the test set.

### 5.2 Experimental Setup

We evaluate our methods on three baselines: BILSTM Sak et al. (2014)and BERT-base Devlin et al. (2018), CDN-Baseline(from CBLUE)Zhang et al. (2021). For BILSTM, we use two BILSTM layers followed by a MLP layer to perform classification. For BERT-based models, we use the CLS vector to perform classification. For CDN-Baseline, we use the original model provided by its git repository[2], which follows a "recall-match" two step training approach based on pre-trained language models. The choose of the baseline models is to demonstrate the effectiveness of our method under different types of models and training

---

[2]https://github.com/CBLUEbenchmark/CBLUE

Table 1: Comparison of the devset accuracy (%) for the choice of different data augmentation methods on various baseline models.

| Model | BILSTM | BERT-base | CDN-Baseline |
|---|---|---|---|
| trainset | 0.455 | 0.558 | 0.577 |
| trainset+EDA | 0.451 | 0.519 | 0.561 |
| trainset+BT | 0.466 | 0.556 | 0.578 |
| trainset+DDA | 0.485 | 0.578 | 0.592 |

settings. In specific, we verify the effectiveness of DDA to a train-from-scratch model using a BILSTM model, we verify the effectiveness to models with pre-trained knowledge using the BERT-base model, and we verify the effectiveness to complex models using CDN-Baseline model.

For the BILSTM model and BERT-base model, we use accuracy to judge the model performance. In our evaluation, we treat this disease normalization as a multi-class classification rather than multi-label classification task despite that there are few data samples that a single unnormalized disease is matched to several standard diseases. Hence, if an unnormalized disease is matched to several standard diseases, this data sample is considered correctly predicted as long as one of the standard diseases is correctly predicted. We design the experiments in this way to simplify the model as much as possible to more clearly illustrate the effectiveness of DDA. For CDN-Baseline, we stick to the settings in CBLUE Zhang et al. (2021), which use the F1 as the evaluation metric, use BERT-base as the baseline model, and use the two step training paradigm provided by CBLUE for better comparison.

To ensure fairness, we use the exact same parameter settings for the same model. In particular, for CDN-Baseline, we use almost the same parameter settings as CBLUE's git repository, including random seed numbers.

Additionally, we use devset for performance comparison, since the label of test set of the CHIP-CDN dataset is not given. For all experiments, we keep the best performing result as the final score.

### 5.3 Results

The results are shown in Table 1. The trainset in the table represents CHIP-CDN training set. From top to bottom, the performance of different models using different data augmentation methods is represented. Among them, BT is the back-translation data augment method[3], and DDA is the semantic-based disease name data augmentation method proposed by us. The experimental results demonstrate that although EDA and back-translation increase diversity, they both hurt performances in some settings (especially for EDA). However, DDA improves the performance in every settings. Clearly, DDA avoids the problem of EDA, and its effect is much better than BT.

We observe that the performances improve for all models above after applying the DDA methods, showing the effectiveness of our proposed methods. For the BILSTM model, the relative performance improvement reaches 6%. We further observe that there is more performance gain on BILSTM than BERT-based models and CDN-Baseline, probably because the knowledge in pre-trained language models has already covered some of the similar information, but our proposed method can further improve their performance, showing the effectiveness of DDA.

### 5.4 Ablation Study

In this section, we evaluate the effectiveness of every data augmentation methods on BILSTM, BERT-base models and CDN-Baseline. As we propose two types of data augmentation methods, we evaluate them by taking out these methods one by one to see the resulting performances. The results are shown in Table 2. We observe that removing data gener-

---

[3]we use the youdao translation tool and the URL is https://fanyi.youdao.com/.

Table 2: Ablation study for CHIP-CDN dataset. We remove our proposed data augmentation methods once at a time and evaluate the results.

| Strategy | BILSTM | BERT-base | CDN-Baseline |
|---|---|---|---|
| DDA full | 0.485 | 0.578 | 0.592 |
| - AR | 0.467 | 0.568 | 0.588 |
| - MGA | 0.455 | 0.558 | 0.577 |

ated by either types of methods would lead to performance degradation, thus proving the effectiveness of every method that we propose.

## 5.5 Smaller datasets experiments

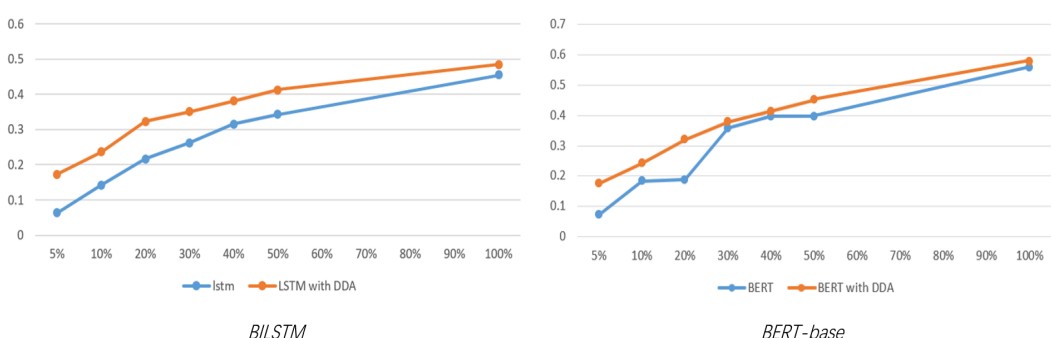

| BILSTM | BERT-base |
|---|---|

Figure 3: Performance comparison on smaller datasets for BILSTM and BERT-base. The smaller datasets are derived by randomly sampling the original CHIP-CDN training set, and the devset of CHIP-CDN stays the same.

We also evaluate the performance improvements over smaller datasets that derived from CHIP-CDN since the data scarcity problem is more severe in smaller datasets. We evaluate the training set whose sizes range from 5%, to 100% of the CHIP-CDN training set size. For the convenience of training, for augmented training in this setting, we only leverage standard disease names in ICD-coding. No data from disease normalization training set are used.

We draw curves to illustrate the comparison on whether to use our proposed methods or not, which is shown in figure 3. When the size of the training set increase, both curves steadily improve. We also notice that the performance gain is higher when the size of the training set is smaller.

## 6 Conclusion

In this paper, we propose two main types of data augmentation methods for Chinese disease normalization tasks based on two hypothesis respectively, where the disease names have the property of structural invariance, and the labels in disease normalization task have the transitivity properties. Our data augmentation methods explore the semantic information and the relation information in diseases, and are adopted in augmented training fashion to avoid introducing misinformation. Experimental results show that our DDA method can better solve the three main challenges in disease normalization task, namely description diversity, data scarcity, and semantics density. Compared to EDA and back-translation methods, our method has obvious advantages on the disease normalization task. Furthermore, we prove that our data augmentation methods work even better on smaller datasets.

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

Table 3: Statistics of the resulting augmented data.

| Method | AR1 | AR2 | AR3 | AR4 | AR5 | AR6 | MGA1 | MGA2 | MGA3 |
|--------|-----|-----|-----|-----|-----|-----|------|------|------|
| num | 443916 | 202587 | 98130 | 360900 | 18373 | 56431 | 23275 | 8870 | 3045 |

Table 4: Hyperparameter settings for all three baseline models

| Model | stage | learning rate | epoch |
|-------|-------|---------------|-------|
| BILSTM | augmented training | 1e-4 | 100 |
| BILSTM | fine-tune | 1e-4 | 100 |
| BERT | augmented training | 1e-5 | 1 |
| BERT | fine-tune | 1e-4 | 5 |
| CDN-Baseline | augmented training | 5e-6 | 1 |
| CDN-Baseline | fine-tune | 5e-5 | 3 |

# A  Appendix

## A.1  data augment result statics

The table 3 is all the statistical results of the data we obtained using MGA and AR data augmentation methods [4].

## A.2  Hyperparameter settings

Table 4 shows the hyperparameter settings of our choices. For different methods, the way of parameter setting is different. For models that use word2vec initialization or random initialization parameters, the training on augmented data can be regarded as a special pre-training task, and a large learning rate and a large number of iterations can be set to make the training sufficient. For models that use a pre-trained model (i.e. BERT) as the backbone, a small learning rate and a small number of training iterations should be set to avoid the catastrophic forgetting of valuable information in the pre-trained models.

For each baseline model, we first train on the augmented dataset (Augmented Training), and then fine-tune on CHIP-CDN dataset. For the CDN-Baseline model, we use Chinese-bert-wwm as the pre-training model, and the training method is provided by CBLUE. For the DDA method, we first use the augmented dataset to train for 1 epoch with a learning rate of 5e-6 and then fine-tune on CHIP-CDN. The hyperparameter of the num_negative_sample is 3+3 and the recall_k is 2 (The explanation of hyperparameter num_negative_sample and recall_k can be found in their github repository).

## A.3  Analysis

In table 5, the first row represents the distribution of the number of times the label appears in the training set. The other two rows represent the label distribution of the two types of the augmented data. The statistical result shows that the data of DDA can effectively improve the labels that appear less frequently (the number of times < 3) and the labels that do not appear at all (did not appear) in the training set. This is beneficial for addressing the data scarcity problem of disease normalization tasks and the diversity of disease names. This is the direct reason why DDA works. As for EDA and BT, they can only increase the number of labels that are already appeared in the training set, which only solve the problem of expression diversity. Hence, their abilities are limited.

## A.4  Case Study

We give a real example of the augmentation results generated by different data augmentation methods. We observe that the semantic meaning of the EDA-generated result dramatically

---

[4]We will open source the augmentation code and the augmented result on Github.

Table 5: Analysis the label distribution of DDA result.

| Data | all | did not appear | occurs<=3 | occurs>3 |
|---|---|---|---|---|
| trainset | 37645 | 34396 | 2870 | 634 |
| DDA-AR | 11518 | 9532 | 1497 | 489 |
| DDA-MGA | 8478 | 7315 | 915 | 246 |

Table 6: Case Study, Compare the generated result through three data augmentation methods.

| Model | text | label |
|---|---|---|
| original | 踝关节创伤性关节病变 | 踝关节损伤 |
| - | Traumatic Arthropathy of the Ankle | ankle injury |
| EDA | 创伤性关节病变 | 踝关节损伤 |
| - | traumatic joint disease | ankle injury |
| BT | 创伤性踝关节损伤 | 踝关节损伤 |
| - | Traumatic Ankle Injury | ankle injury |
| DA-AR | 膝关节创伤性关节病变 | 膝关节损伤 |
| - | Traumatic Arthropathy of the Knee | knee injury knee injury |
| DA-MGA | 踝关节创伤性关节病变 | 踝扭伤和劳损 |
| - | Traumatic Arthropathy of the Ankle | Ankle Sprains and Strain Injuries |

changes due to the property of semantics density, and it changes the key information within the disease by losing the anatomical location. The results generated by BT is more realistic, but this method cannot generate samples beyond the original label scope, and it also suffers from the restrictions of the translation tools. As for our proposed method DDA (last two lines in the table), it can not only increase the diversity of the input, but also generates data where their labels are never appeared in the training set, so that sparse labels can be trained more thoroughly.

A.5   Future work

So far, we have only proved the effectiveness of our DDA method, but no experimental analysis is done to explore the internal mechanisms of why it is so effective. Moreover, to further avoid the injection of misinformation, we believe designing loss function terms to effectively select more valuable data from the data augmentation results can be a promising direction. We aim to perform researches on those topics in the future.

