# OpenReview forum: "Exploring semantic information in disease: Simple Data Augmentation Techniques for Chinese Disease Normalization"
_ICLR.cc/2023/Conference — Submitted to ICLR 2023_

### Official Review · Reviewer_Ujvh · 2022-10-21

**Confidence:** 3
**Correctness:** 2
**Technical Novelty And Significance:** 3
**Empirical Novelty And Significance:** 2
**Recommendation:** 3

**Clarity, Quality, Novelty And Reproducibility:**

The paper seems a little bit redundant, e.g. two assumptions (structural invariance, transitivity) are repeated by many times.  -- it seems the naming of such two properties is not that appropriate.

Questions:
- Why don't the authors also benchmark the DA method in English disease normalization dataset (if there is any). From my understanding, some of the DA methods are specific to Chinese language, right? If yes, this should be clearly stated. Otherwise English disease normalization dataset should be considered.

- Do you think it is more challenging for Chinese language than English to deal with  disease normalization dataset?  If so, please illustrate a little bit morel,


**Strength And Weaknesses:**

## Strength:
- The paper proposes an interesting method to augment dataset, it achieave good performance compared with other augmentation method.
- It demonstrated that EDA and BT methods are harmful  on CDN datasets.

## Weakness:
- It would be nice if the authors could basically check how many ratios of  wrong labels brought by the proposed DA and existing one, either using expert annotation  or automatic evaluation. This could be tested by using only a few examples.

- There are no details about how many augmented examples used for baseline DA methods including EDA and BT. This should be carefully compared for a fair comparison.

- Only a small-scaled Chinese dataset was used. It would be better if the authors could evidences their method across many datasets

- The task is only for Chinese language, making it too narrow to fit ICLR.

- The authors should detail the motivaiton to use EDA and BT as the baselines. Since there are many other  DA methods in NLP, see https://arxiv.org/abs/2110.01852




## minor issue:
- The font and the reference format seems wrong

**Summary Of The Paper:**

 This paper introduces a new data augmentation method for Chinese disease normalization dataset after analysing the unnormalized disease name and standard disease name. The main contribution is a novel data augmentation method adapted to a new dataset, which consists of axis-word replacement and multi-grain aggregation.



**Summary Of The Review:**

The paper focus a intereting problem although the problem seems not a core problem in ICLR community. The method seems sound but I think it is not properly compared with various existing DA methods

---

> ### Author Response · Authors · 2022-11-15
> **Response from the authors**
>
> Dear distinguished reviewer, thank you so much for your review. I will try my best to answer your questions.
>
> The amount of enhancement data used in training is indeed an important issue that we have ignored. The DDA method did not use all the enhanced data during training. For methods such as AR1, we only randomly sampled some data due to the large number of generated data, while for methods such as MGA3, we used all generated data. Finally, we used about 300000 data for training. For EDA and BT methods, we only used about 100000 data. Because we found that with the increase of data volume, EDA and BT methods cannot continuously increase the model performance. Therefore, although the data volume is different, the comparison is still fair. We will add this content in the paper.
>
> The advice on evaluating the methods across many datasets is very good, but unfortunately we only found one publicly available Chinese disease normalization dataset. We believe the work would be more convincible if we stick to public dataset.
>
> For baseline data augmentation methods, we chose EDA and BT because they are classical and universal data augmentation methods that are adopted in NLP. Although there are other baselines, many of them are not suitable to compare with our method because their targeting areas are a bit narrow (see our paper section 3.1).
>
> As for whether our DA methods are specific to Chinese language, we believe it is not, but it will be more difficult when applying them to English language, especially the axis replacement method, as a single English word may have different forms (e.g., noun, adj.), so it should be more careful when replacing words.

---

### Official Review · Reviewer_6zpd · 2022-10-24

**Confidence:** 5
**Correctness:** 1
**Technical Novelty And Significance:** 2
**Empirical Novelty And Significance:** 2
**Recommendation:** 1

**Clarity, Quality, Novelty And Reproducibility:**

The paper addressed a fundamental task in medical natural language processing: disease name normalization. However, although this task is extensively studied, also in Chinese, the paper did not seem to properly acknowledge and appreciate this prior work. To illustrate, its first two sections (Introduction and Background) did not use peer-reviewed publications to support its argumentation (in total, one paper (arXiv preprint from 2019) was cited in the entire first section and none of the related literature was cited in the second section) and this problem of insufficient positioning of this paper in the prior work was also evident in the last section (Conclusion) of the paper; no discussion section or paragraph to position the contributions of this paper compared to the prior work was included here to close the study. Finally, the third section (Related Work) was hard to understand because the scope of the included studies was not communicated; I would have wanted to learn if, for example, the included papers were targeting or at least applicable to medical natural language processing in Chinese. Consequently, the novelty and clarity of the study - or minimally their clear and convincing argumentation - can be questioned.

Given the societal importance of medical natural language processing, I would have expected to see a broader impact statement in this paper. Also, although the CHIP-CDN - Chinese disease normalization dataset used in this study seems to have been collected and released before, the authors should have discussed relevant ethical considerations, minimally describing how the authors analyzed and evaluated that the dataset was created ethically and that their use of its for the purposes of this study was ethical.

Unfortunately, the paper was formatted using an incorrect template; it did not look like ICLR 2023 papers and hence, should be desk rejected. Also, further care should have been demonstrated in writing; for instance, the paper title was inconsistently capitalized.

**Details Of Ethics Concerns:**

Given the societal importance of medical natural language processing, I would have expected to see a broader impact statement in this paper. Also, although the CHIP-CDN - Chinese disease normalization dataset used in this study seems to have been collected and released before, the authors should have discussed relevant ethical considerations, minimally describing how the authors analyzed and evaluated that the dataset was created ethically and that their use of its for the purposes of this study was ethical.

**Strength And Weaknesses:**

* Experimental results demonstrated a fair 3% performance gain with potential for further gains on smaller datasets.
* Medical natural language processing is societally important.
* Although the evaluation results demonstrated performance gains, no statistical significance testing, confidence intervals, or effect sizes were reported in the paper. However, the authors reported on an ablation study whose merit should be noted.
* The novelty and clarity of the study - or minimally their clear and convincing argumentation - can be questioned.
* Ethical considerations of this study were insufficiently addressed.
* Unfortunately, the paper was formatted using an incorrect template; it did not look like an ICLR 2023 paper and hence, should be desk rejected.


**Summary Of The Paper:**

The proposed and evaluated data augmentation techniques for normalizing disease names in Chinese. The techniques were derived from disease corpora extracted from an existing dataset of authentic clinical text and the standardized ICD-10 coding system. Experimental results demonstrated a fair 3% performance gain with potential for further gains on smaller datasets.

**Summary Of The Review:**

The paper has more weaknesses than strengths and hence, I recommend its rejection.

---

> ### Author Response · Authors · 2022-11-15
> **Response from the authors**
>
> Dear distinguished reviewer, thank you so much for your review. I will try my best to answer your questions.
>
> Firstly and most importantly, thanks for pointing out the formatting problem. We respect that you think the paper should be desk rejected. However, we did use the ICLR 2023 template. The slight difference in the format is due to the inclusion of Chinese words. Regarding this issue, we have discussed with the conference committee, and they said this format is fine.
>
> Like other reviewers, we do not think that this paper has ethical problems, because what we are studying is only the name of the disease and does not involve patient information, diagnosis and treatment process. Its research results are only used to better standardize the name of the disease, as an auxiliary tool for medical research. Moreover, we only use public dataset to conduct experiments.
>
> Regarding the related work, they are not directed targeting to Chinese medical entity data augmentation as there are not much of study about it. However, all the ideas of the related works are applicable to our study for they are either targeting at data augmentation or medical data operation. Also, although some of the papers are targeting at English corpus, their idea can transfer to Chinese. For example, Falis et al. (2022) use UMLS to find medical synonyms. This can also be applied to Chinese corpus as long as there is also a database containing synonyms for Chinese medical terms.

---

> > ### Comment · Reviewer_6zpd · 2022-11-21
> > **Acknowledging the author response**
> >
> > I am acknowledging that the authors have responded to my review above. However, their response does not seem to be addressing my concerns. Hence, I am keeping my original review unchanged.

---

### Official Review · Reviewer_ethR · 2022-10-25

**Confidence:** 3
**Correctness:** 2
**Technical Novelty And Significance:** 2
**Empirical Novelty And Significance:** 2
**Recommendation:** 3

**Clarity, Quality, Novelty And Reproducibility:**

Quality: The paper is of reasonable quality.
Clarity: Table 6 is unclear.The comparison between different diagnosis codes goes from ankle to Knee and the label in the DA-AR is duplicated.So, not sure how to assess.Not sure if the challenge is different or even worse as there is a language component in it too(Chinese vs English).
Originality: While the authors show incremental benefit of their technique over prior related work, not sure if the gains can be generalized.

**Strength And Weaknesses:**

Strengths:
- Authors work does demonstrate incremental benefit(3%) over prior related work.
- Clearly,this is valuable works there is significant variability in disease diagnosis as documented by clinicians.

Weakness:
- Incremental benefit is limited.
- Some of the grammar could be improved.
- Unclear if the Chinese language have the exact translation for comparison.

**Summary Of The Paper:**

The paper describes authors work related to set of data augmentation techniques that work together as an augmented training task for disease diagnosis normalization.Various techniques are described which show incremental improvement in disease diagnosis normalization against ICD-10 classification using CHIP-CDN dataset. The key to their work are described methods which augment the labels.

**Summary Of The Review:**

The paper addresses an important challenge faced in healthcare data,especially with disease diagnosis. Abbreviations,different standards, different languages and different coding nomenclature(ICD vs others), make it difficult to find the correct diagnosis in the text. This is a challenge from not just label point of view but also from any other modeling aspect.
The authors have shown through their work that their novel techniques can show incremental benefit in overcoming this challenge.
But it is not clear, whether this is generalizable or not.

---

> ### Author Response · Authors · 2022-11-15
> **Response from the authors**
>
> Dear distinguished reviewer, thank you so much for your review. I will try my best to answer your questions.
>
> Thank you for pointing out the grammar issue. We are working on this and aiming to produce a more readable paper.
>
> We believe the comparison for Chinese-English translation is fair since Chinese and English disease name both contain anatomical location, disease center, etc. Moreover, we believe English disease axis replacement is a bit more difficult than Chinese as a single English word may have different forms (e.g., noun, adj.).
>
> For Table 6 comparison, just compare the DA-DR text with the original text. The only difference is the word Ankle is replaced by Knee.

---

> ### Comment · Reviewer_ethR · 2022-11-27
> **Review and revision of the recommendation**
>
> Responses from the authors to my comments and concerns including explanation of the differences are not satisfactory. For e.g, an ankle injury is very different from knee injury,so the accuracy cannot be judged based on the substitution as suggested by the authors.
> I do not see a substantial explanation to the concerns raised by other reviewers either.
>
> Change my recommendation to 3: reject, not good enough.

---

### Official Review · Reviewer_iwv2 · 2022-11-05

**Confidence:** 5
**Correctness:** 3
**Technical Novelty And Significance:** 2
**Empirical Novelty And Significance:** 2
**Recommendation:** 3

**Clarity, Quality, Novelty And Reproducibility:**

The paper is clearly written and easy to follow but needs some more work for refining the language and presentation. The work is novel but needs some more consideration for the validity of pre-training before the task fine-tuning. Currently, it would be quite hard to reproduce the results.

**Strength And Weaknesses:**

Strength:
1. The authors have focused on an important problem of disease normalization.
2. The proposed framework improves the performance of the existing normalization models.

Weaknesses [Questions to the authors]:
1. The paper needs some proof-reading and formatting. Some common formatting issues are highlighted below:
    - The starting double quotes in latex need to be `` as opposed to " since they appear inverted during pdf built.
    - All the tables would look a bit better with the top and bottom margins would look a bit cleaner.
    - Para 1 of the introduction consists of multiple sentences with grammatical errors as well.
2. The authors have mentioned the work of Falis et al [2022] and Abdollahi et al [2021] for data augmentation using UMLS but have not compared their methodologies for data augmentation.
3. The information regarding the NER tool are completely missing from the paper. On which data was the NER model from section 4.1 trained and what was its performance? Since the performance of the framework heavily depends on the model as well.
4. Is there a validity study done to see if the axis rotation method would not create diseases which are not anatomically or medically correct or significant?
5. Section 4.2 needs some more details regarding the training procedure. Were the experiments done using multiple seeds? If yes, could you please share the average and std for the results?
6. Details in Section 5.2 are confusing. Para 2 of the section mentions accuracy as the evaluation metric but also mentions that F1-score was used for evaluation according to the previous work. Also, was macro or micro F1-score used for evaluation?
7. Table 1 mentioned that the result is of devset accuracy. Are all the experiments performed on devset? And is the testset accuracy or F1-score not reported?

-------------------------------------
Recommendation to the authors
-------------------------------------
I believe that the work is important and necessary but I would also like to add that it would be much more suitable for a clinically oriented workshop or conference such as ML4H, MLHC, LOUHI, BioNLP or clinicalNLP.


**Summary Of The Paper:**

The authors have proposed a data augmentation framework which helps in improving the performance of existing disease normalization models. The framework is compared against two existing methodologies and in smaller dataset setting as well.

**Summary Of The Review:**

Currently, the work needs some more effort on the presentation and proof-reading. The experimentation and training procedure needs to be more clearly written and experiments need to be a bit more thorough in terms of number of runs. Some more baselines would also improve the quality of work.
As I mentioned earlier, I believe that the work is important and necessary but I would be much more suitable for a clinically oriented workshop or conference such as ML4H, MLHC, LOUHI, BioNLP or clinicalNLP.

---

> ### Author Response · Authors · 2022-11-15
> **Response from the authors**
>
> Dear distinguished reviewer, thank you so much for your review. I will try my best to answer your questions.
>
> Answer to Weaknesses 1: Thank you so much for the advice on formatting. We have changed that according to your suggestions.
>
> Answer to Weaknesses 2: For model comparison, we did cite the works of Falis et al [2022] and Abdollahi et al [2021]. We’re sorry for not clarifying this in the paper, but those papers focus on classification on long medical texts where medical terms are small parts buried in it, so they substitute the whole term instead of parts of the term, and that makes our methods very distinct from them. Additionally, those paper use synonyms in English, which is not suitable to directly apply them on Chinese corpus. The reason for not suitable for comparison is included in the paper now.
>
> Answer to Weaknesses 3: We are sorry that we didn't clarify this issue. The NER model we use is trained from manually annotated data. We invited doctors to mark 5000 disease names and divide them into various elements as described in the paper, such as human body part, nature, etc. We divide it into training set and test set according to 4:1, and use the traditional BERT+CRF model for training. The test results showed that the F1 score of this model on the test set is 91.3%. We will include these details in our paper.
>
> Answer to Weaknesses 4: For the validity study to see if the generated diseases are authentic, we argue that for this specific task the model does not always need disease names to be 100% correct. As long as the model can learn enough semantic information buried in disease names, it should improve the model’s performance.
>
> Answer to Weaknesses 6: The original competition of the CHIP-CDN dataset treats this normalization task as a multi-label classification task, and the F1 score is their own definition, which is now illustrated in the paper. However, we found most of the data only contains one label, so to simplify the problem we just treat this task as single-label classification and use accuracy to evaluate the model as this will give us more insight about how well the model is benefited from our proposed data augmentation method.
>
> Answer to Weaknesses 7: As indicated in the paper (Section 5.2 last paragraph), all the experiments are evaluated on devset as the testset of this dataset is not given.

---

### Decision · Program_Chairs · 2023-01-20

**Decision:**

Reject

**Justification For Why Not Higher Score:**

Lack of novelty

**Justification For Why Not Lower Score:**

no lower score possible

**Metareview: Summary, Strengths And Weaknesses:**

SUMMARY

The paper proposes and evaluates data augmentation
techniques for normalizing disease names in Chinese. The
data used are derived from disease corpora extracted from
ICD-10 and clinical text.  Experiments demonstrate a 3%
performance gain with potential for further gains on smaller
datasets.

STRENGTHS

Important problem

Improved performance compared to prior work

Helpful ablation study

Linguistic contribution: the problem of normalization has
not been studied as extensively for Chinese as for English

WEAKNESSES

The work is rather incremental and lacks novelty.

The paper should explain more clearly how normalization
differs in English and Chinese (i.e., why it's not just a
matter of applying existing English techniques to Chinese).

Lack of comparison to important baselines

Experiments not described in sufficient detail, e.g.,
NER. Partially, but not sufficiently addressed in the response.

Experimental results vary depending on seed, but variation/confidence is not measured.

Evaluation on dev set only, lack of test set

Small-scale evaluation

A qualitative manual evaluation on a small sample is highly desirable.

Writeup is unclear and has poor quality.